# Development of a Chicken Gastrointestinal Tract (GIT) Simulation Model: Impact of Cecal Inoculum Storage Preservation Conditions

Nelson Mota de Carvalho [1], Célia Maria Costa [1], Cláudia Castro [1], Mayra Anton Dib Saleh [1], Manuela Estevez Pintado [1], Diana Luazi Oliveira [2] and Ana Raquel Madureira [1,*]

[1]  CBQF—Centro de Biotecnologia e Química Fina—Laboratório Associado, Escola Superior de Biotecnologia, Universidade Católica Portuguesa, Rua Diogo Botelho 1327, 4169-005 Porto, Portugal; ncarvalho@ucp.pt (N.M.d.C.); cfcosta@ucp.pt (C.M.C.); cmocastro@ucp.pt (C.C.); mayra.ad.saleh@uac.pt (M.A.D.S.); mpintado@ucp.pt (M.E.P.)
[2]  Research and Innovation Unit—Instituto de Investigação e Inovação em Saúde, Universidade do Porto, Rua Alfredo Allen, 208, 4200-135 Porto, Portugal; dianao@i3s.up.pt
*   Correspondence: rmadureira@ucp.pt

**Abstract:** A chicken gastrointestinal tract (GIT) simulation model was developed to help predict the potential effects of feed additives supplementation on chicken' microbiota. The chemical and enzymatic conditions for oral, gastric, intestinal, and cecum fermentation phases were designed to closely resemble the chicken GIT conditions. For cecum fermentation, the inoculum was obtained from the cecal contents of 18 38-day broiler chickens. The impact of inoculum preservation on bacteria viability was assessed by comparing two methods of preservation with fresh inoculum: (1) 5% dimethyl sulfoxide (DMSO) at −80 °C and (2) 30% glycerol at −20 °C. The fermentation with fresh and frozen (DMSO method) inoculums was performed and compared using standard chicken feed (SCF) and SCF with 1% fructooligosaccharides (FOS), and inoculum control (IC) condition without feed matrix was used as a baseline. Inoculum's viability was assessed throughout 90 days of storage by culture media platting, while bacterial growth and metabolites production during fermentation was evaluated by quantitative polymerase chain reaction (qPCR), high-performance liquid chromatography (HPLC), and total ammonia nitrogen quantification. The DMSO method was shown to be the most suitable for cecal inoculum storage. Higher growth of beneficial cecal bacteria for fresh inoculum was observed in SCF while for frozen inoculum, was the SCF + FOS condition. Also, frozen inoculum had lower activity of butyrate producers and proteolytic bacteria, showing different fermentation profiles. The GIT model developed showed to be useful to test the effect of feed additives supplementation.

**Keywords:** chicken; *in vitro* model; gastrointestinal tract; preservation; viability; fermentation; cecal microbiota; organic acids; ammonia





## 1. Introduction

With the estimated global population reaching 9.6 billion people by 2050 and the increasing demand for animal and meat products, there is a need for animal husbandry and food production systems to increase their production level by 60 to 110% by 2050 [1,2]. Over the last decades, the poultry industry contributed significantly to the human diet worldwide, by providing protein sources, i.e., meat and eggs [3]. Worldwide, poultry production has been growing rapidly to the point that the increasing demand for chicken meat has driven livestock production systems to genetically select chickens with fast growth performance that can achieve a live weight of 2 to 3.2 kg in 35–45 days (slaughter age) [4–7].

One of the biggest production costs in the poultry industry is related to animal feed, which corresponds to 60–75% of the total production cost [1,8,9]. Strategies to maximize economic viability and high-quality products are mandatory to ensure a cost-efficient industry,

in which farming, poultry health and performance, and high-quality meat production are key elements. Currently, animal nutrition is a crucial factor in the poultry industry economy, resourcing to an optimized dietary practice, taking into account the animal's physiology and the impact of the dietary components on their performance, health, and ultimately on the delivery of high-quality products [8,10]. For that, it is crucial to understand the chicken's nutrient and metabolic needs, their physiology, and the impact that external factors may have on their health and well-being [1,11].

Feed additives play an important role in regulating intestinal microbiota, promoting the animal's general health and well-being, and becoming a good replacer of the antibiotic growth promoters (AGP), the use of which has been progressively limited and even forbidden in several countries due to their toxicity (for animals and humans), the rise in antibiotic-resistant microorganisms, and the increment of the host's susceptibility with the prevalence of intestinal pathogens [3,12–14]. Prebiotics have been often suggested as healthier, less harmful, and more sustainable alternatives to antibiotics and synthetic additives, to proactively prevent health issues and provide supplementary nutrients and additional health benefits [13]. Numerous studies have been proving the benefits of adding prebiotics to feed matrices, both in terms of nutrition and health [15–17]. The most recognized prebiotics for poultry nutrition are inulin and FOS, which are substrates selectively consumed by the host microorganisms, conferring health benefits [18].

Most of the studies that assess the impact of prebiotic additives in poultry focus on the modulation of the cecum microbiota, since this region, in terms of microorganisms, is the most populated and diverse section of the poultry GIT and the main region of bacterial fermentation [19,20]. The use of GIT *in vitro* models (mimicking the oral phase, gastric-intestinal digestion phase, and intestinal absorption phase) coupled with *in vitro* fermentation system (mimicking the cecum fermentation phase) allows for the simulation of the physicochemical and physiological events on the digestive tract mirroring the structural changes, feed digestibility, and fermentation by cecum microbiota, predicting the potential *in vivo* effects without resort to animal trials in such preliminary stages [21]. Cecum fermentation simulation models usually use cecum contents as inoculum to replicate the cecum microbiota and simulate the fermentation, as these inoculums are representative of such ecosystems [19,22–24]. Cecum microbiota has a direct impact on metabolic processes that influence the animal's growth and performance. Moreover, the different microbial species are responsible for the production of metabolites, such as short-chain fatty acids (SCFA) and ammonia which are closely associated with the host's health, as reviewed in several studies [10,25,26].

An important aspect related to *in vitro* fermentation studies is the physical–chemical characteristics of the inoculum. One clear example that can affect the inoculum viability and fermentability is the storage time at temperatures below 0 °C [27]. Therefore, one of the strategies to minimize the impact of freezing the inoculum is the use of cryoprotectants such as DMSO and glycerol, enabling to have reproducible, robust, and reliable *in vitro* fermentation experiments [27].

The aim of this study is to provide a practical and trustworthy chicken GIT *in vitro* model as a tool for the preliminary assessment of the potential of novel and well-established ingredients, developed for poultry feed avoiding excessive and unnecessary animal *in vivo* studies. To reach this aim, the authors: (1) developed a chicken GIT simulation model with oral, gastric, intestinal, and cecum fermentation stages; (2) compared and evaluated two preservation methods to maintain the bacterial viability of the cecal inoculum for up to 90 days; and (3) compared the *in vitro* fermentation profiles of the fresh inoculum with a best-preserved inoculum of the two methods tested, in terms of bacterial growth, organic acids, and ammonia production.

## 2. Materials and Methods

### 2.1. Reagents/Chemicals and Apparatus

All reagents/chemicals and equipment used in this study are described in Appendix A.

## 2.2. Study Experimental Design and Objectives

A summary of the experimental methodology is presented in Figure 1 and a detailed description of the methods/techniques used throughout these studies are presented in the sub-sections below: Section 2.3—Bacterial viability study and Section 2.4—Cecum microbiota assessment study.

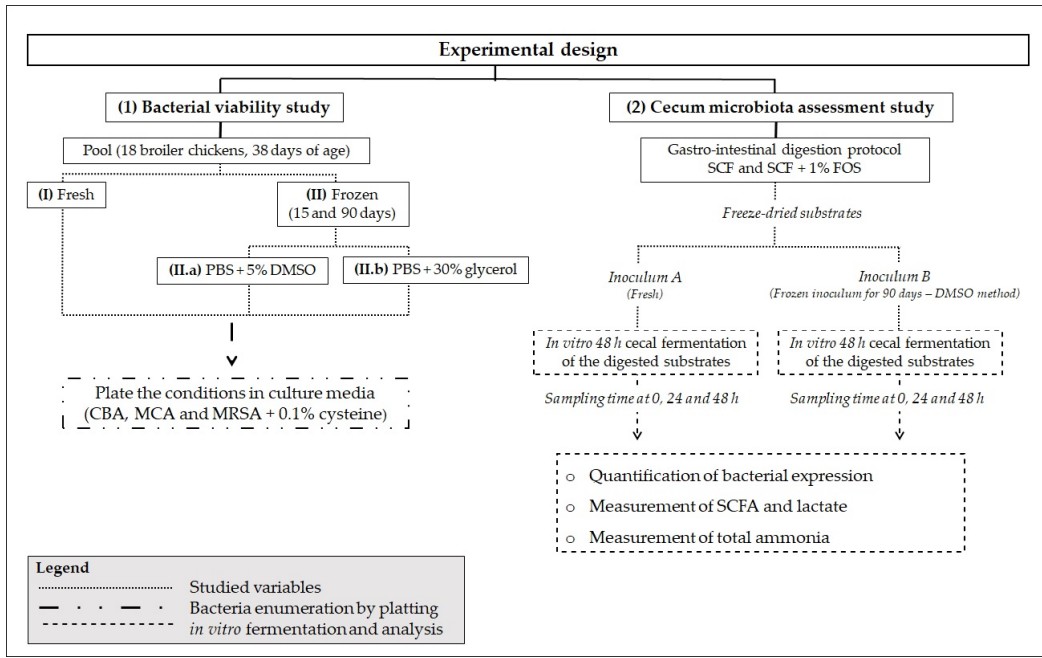

**Figure 1.** Schematic model of the methodology applied in this study. CBA (Columbia blood agar); MCA (MacConkey agar); MRSA (de Man, Rogosa, and Sharpe agar); PBS (phosphate-buffered saline).

The bacterial viability study was performed with the intention to understand which preservation method would be more recommendable to store for a long period of time (i.e., 90 days) the cecal inoculum that will be used on the chicken GIT model developed.

The cecum microbiota assessment study was performed to develop a chicken GIT model and to compare the cecum fermentation parameters (i.e., bacteria, organic acids, and ammonia) of using the same inoculum in two different states: Fresh inoculum (on the day of preparation) and frozen inoculum (90 days stored on a well-preserved method).

## 2.3. Bacterial Viability Study

### 2.3.1. Fresh and Frozen Cecal Inoculum Preparation

Cecal content was obtained from 18 carcasses of 38-day-old Ross 308 broilers from the same production system, with an average weight of 1.91 ± 0.19 kg at the Savinor slaughterhouse (Covelas, Portugal), fed an antibiotic-free maize and soybean diet (Table S1). These broiler chickens were raised in a production system with the intention to use these animals for human and/or animal consumption. Before their slaughter, broiler chickens underwent a fasting period of 8–10 h. The birds were subjected to electrical stunning, and the cecum was removed, clipped on both sides with a string, properly identified, and stored in clean tamper-proof specimen 1 L containers. The containers were placed in a 2.5 L anaerobic jar containing an anaerobic generation sachet, closed, and only opened inside in an anaerobic cabinet (nitrogen 80%, carbon dioxide 10%, and hydrogen 10%), within 2 h of collection.

Under anaerobic conditions, the cecal content for each cecum was squeezed into an empty pre-weighted tamper-proof specimen 1 L container and weighted. I cecal pool was diluted at 10% (*w/w*) with 0.1 M PBS solution for the fresh inoculum (I—inoculum A), one of the frozen inoculums (IIa.—inoculum B) was diluted at 10% (*w/w*) 0.1 M PBS solution with 5% DMSO, and the other frozen inoculum (IIb.—inoculum C) was diluted

at 10% (*w/w*) 0.1 M PBS solution with 30% glycerol. Inoculum B was stored at −80 °C while inoculum C was stored at −20 °C. Inoculum C was prepared with the intention of comparing the preservation method tested in inoculum B with a preservation method previously published by the authors [27]. The cecal slurry was first homogenized manually and further mechanically with a Mixwel® laboratory blender (Alliance Bio Expertise, Guipry, France) for 2 min at 460 paddles beats/min and aliquots of 15 mL cecal inoculum were prepared. Cecal bacteria viability of the pooled inoculum was assessed on the same day of preparation, and 15 and 90 days after storage at −80 °C using culture-dependent methods, as described in Section 2.3.2. Figure 2 presents the methodology used to assess the cecal inoculum viability.

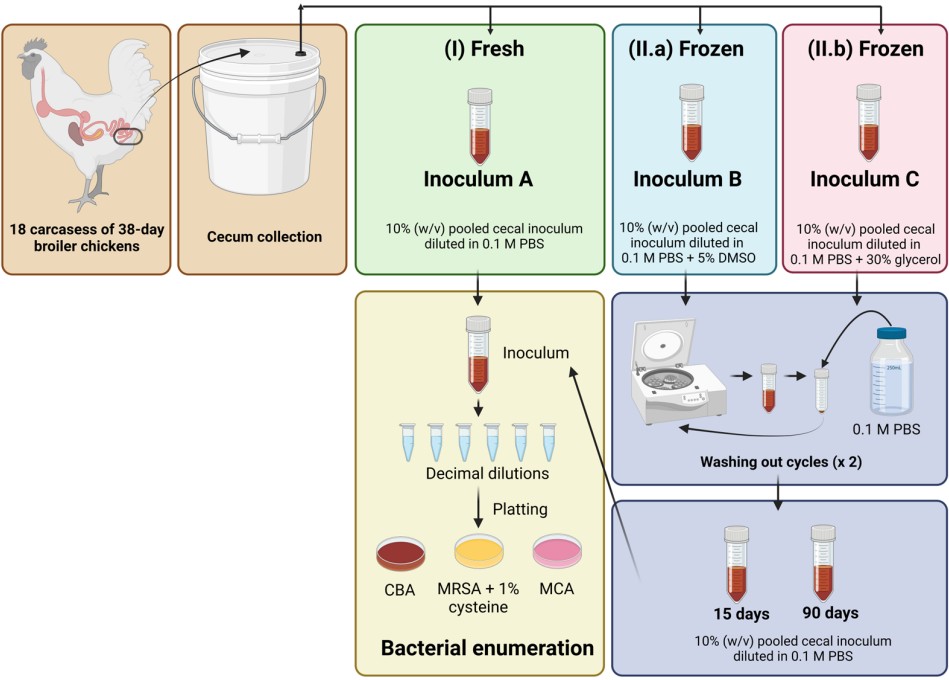

**Figure 2.** Chicken cecal inoculum bacterial viability study work flowchart related to the fresh and frozen cecal inoculum used.

### 2.3.2. Bacterial Enumeration by Culture-Dependent Methods

The cecal inoculums A (on the day of preparation), B (15 and 90 days of storage at −80 °C), and C (15 and 90 days of storage at −20 °C) were diluted in 0.1% (*w/v*) peptone water (decimal dilutions) and platted in different culture media, as described in Table 1, using the Miles and Misra technique [28]. Platting was performed in anaerobic conditions at 37 °C.

**Table 1.** Bacterial enumeration platting conditions of the fresh cecal inoculums.

| Culture Media | Incubation Conditions | Target Bacterial Groups |
|---|---|---|
| CBA with 5% (*v/v*) defibrinated sheep blood (DSB) | Anaerobic 37 °C for 72 h | Total anaerobic bacteria |
| MCA | | Gram-negative and enteric bacteria |
| MRSA with 0.1% (*w/v*) cysteine | | Lactic acid bacteria (LAB) and Bifidobacterium |

*2.4. Cecum Microbiota Assessment*

2.4.1. SCF

In this study, the SCF used was kindly provided by Sorgal S.A. (Aveiro, Portugal). The provided SCF is used as a complete feed, in mash form, for broiler chickens from 23 days old until slaughter. This SCF is composed of corn, peeled, and roasted soybean bagasse obtained by extraction, roasted soybean seeds, rapeseed bagasse, poultry fat, calcium carbonate, monobasic calcium phosphate, sodium chloride, and sodium bicarbonate. The composition of SCF is shown in Table 2.

**Table 2.** Composition of the SCF provided by Sorgal S.A.

| | | |
|---|---|---|
| **Analytical constituents (%)** | Protein | 19.40 |
| | Fat | 5.50 |
| | Fiber | 2.60 |
| | Ash | 4.10 |
| | Calcium | 0.80 |
| | Phosphorous | 0.60 |
| | Sodium | 0.15 |
| | Lysine | 1.15 |
| | Methionine | 0.50 |
| **Additives (per kg)** | Vitamin A | 12,000 IU |
| | Vitamin D3 | 2000 IU |
| | Vitamin E (All-rac-$\alpha$-tocopheryl acetate) | 12 mg |
| | Biotin | 0.09 mg |
| | Potassium iodide | 0.70 mg |
| | Copper | 8.00 mg |
| | Manganese | 100 mg |
| | Zinc | 60 mg |
| | Selenium | 0.20 mg |
| | Iron | 21.30 mg |
| | Canthaxanthin | 4 mg |
| | Ethyl ester of beta-apo-8′ carotenoic acid | 14 mg |
| **Coccidiostats and histomonostats (per kg)** | Narasine | 70 mg |
| **Amino acids and analogs (per kg)** | Hydroxy analogue of methionine | 1 g |
| **Digestibility enhancer (per kg)** | Endo-1,4-beta-xylanase | 40 U |
| | Endo-1,3(4)-beta-glucanase | 35 U |
| | Endo-1,4-beta-glucanase | 135 U |
| **Antioxidants (per kg)** | Butylated hydroxytoluene (BHT) | 120 mg |
| **Anti-caking (per kg)** | Sepiolite | 0.20 g |

2.4.2. Chicken GIT Simulation Model

An *in vitro* GIT simulation model according to de Carvalho et al., 2022 [29], adapted from Meimandipour et al. (2009), Martinez–Haro et al. (2009), and Bean et al. (2016), was carried out [22,30,31]. Two different feeds were subjected to this protocol: (1) SCF and (2) SCF supplemented with 1% (*w/w*) FOS (a common percentage of prebiotic additive used in feed additives [17,32]).

Oral Phase

For the oral phase, 208 μL of 1 mM CaCl$_2$ (pH 4.5–5.5) was added to 4 g of the testing condition in a 100 mL Erlenmeyer. The feed mixture was placed at 41 °C for 40 min without agitation.

Gastric Phase

After the oral phase, to mimic gastric digestion, it was added 25 mL of chicken gizzard digestive juice (1 M NaCl, 10 g/L pepsin from porcine gastric mucosa powder, adjusted to pH 2.0 ± 0.1 with 6 M HCl) to the feed mixture and incubated at 41 °C with moderate agitation for 1 h.

Intestinal Phase

After the gastric phase, to initiate the intestinal digestion phase, the feed mixture pH was adjusted to 6.2 using a 1 M $NaHCO_3$ solution, and 2.52 mL of the volume of each condition was discarded and replaced by the same volume with intestinal digestive juice (3.5% (*w*/*v*) bile extract and 0.35% (*w*/*v*) pancreatin from the porcine pancreas in deionized water, pH = 6.80 ± 0.04). The mixture was incubated at 41 °C, moderate agitation for 3 h. The last stage of the GIT is the intestinal absorption phase, in which the simulated feed digest was transferred into a 1 kDa dialysis membrane clipped on both edges, submerged in a 10 mM NaCl solution, and left overnight, stirred at moderate agitation and room temperature [33]. The retained substrates were freeze-dried and used in the *in vitro* batch culture fermentation systems as feed substrates for the cecum fermentation.

Cecum Fermentation

Two sets of experiments with six independent fermentation vessels per set were carried out. Set 1 used fresh pooled cecal inoculum (inoculum A) collected and prepared on the same day and set 2 used frozen pooled cecal inoculum (inoculum B) which was stored at −80 °C for 90 days in the cryoprotectant solution (i.e., 0.1 M PBS + 5% (*v*/*v*) DMSO) after collection. Before the use of the frozen inoculums in the fermentation, these inoculums were previously submitted to two-DMSO washing-out cycles according to de Carvalho et al. (2021) [27], to eliminate the DMSO, used as a cryo-preservative during storage. Sterile stirred batch culture fermentation vessels of 300 mL were set up and aseptically filled with 135 mL sterile basal nutrient medium according to de Carvalho et al. (2019) [34] and gassed overnight with $O_2$-free $N_2$, with continuous agitation. Each condition was assessed in duplicate, the substrates added aseptically (by flaming the entry/sampling port) and fermented by the cecal inoculums. Vessels (1) and (2) contained 1% (*w*/*v*) of the digested SCF; vessels (3) and (4) 1% (*w*/*v*) of the digested SCF supplemented with 1% (*w*/*w*) FOS (positive control); and in vessels (5) and (6), the IC with no substrate was added (negative control). Once the substrates were properly mixed with the basal media, each vessel was inoculated with 15 mL of fresh or frozen cecal inoculum. A FerMac 260 pH controller was used to maintain the pH between 6.0 and 7.0 (the pH of the chicken cecum) in each vessel [35], and the temperature was kept at 41 °C for 48 h with the help of a water bath. Samples of 10 mL were taken aseptically from each vessel, at 0, 24, and 48 h, and processed according Carvalho et al. (2022) [36]. The analysis of SCFA and lactate was performed by HPLC. The ammonium ($NH_4^+$) concentration was carried out using an ion-selective electrode 9663 of ammonium and the bacterial enumeration was assessed by qPCR (Figure 3).

2.4.3. Bacterial Enumeration by Culture-Independent Methods

The DNA samples from sampling times 0, 24, and 48 h of cecal fermentation were extracted. The total DNA was extracted from the pellet obtained at each sampling time using PureLink^TM Microbiome DNA Purification Kit (Thermo Fisher Scientific, Waltham, MA, USA) according to the DNA extraction protocol provided by the manufacturer. DNA concentration was quantified by a Qubit 4 fluorometer (Thermo Fisher Scientific, Waltham, MA, USA) following the Qubit® dsDNA HS assay kit (Thermo Fisher Scientific, Waltham, MA, USA) protocol. The final DNA concentration of each sample was adjusted to 10 ng/μL. The targeted groups, primer sequences, amplicon sizes, and literature references are depicted in Table 3. Conditions for qPCR reactions were prepared to a final volume of 10 μL, containing 1 × NZYSupreme qPCR Green Master Mix (2×), 1 μM of each primer (forward

and reverse), 2 µL of DNase/RNase-free water, and 1 µL of template DNA. In the negative control, and 1 µL of DNase/RNase-free water was used instead of template DNA. The cycling conditions were 95 °C for 10 min (polymerase activation), 95 °C for 15 s (denaturation), then 40 cycles of 60 °C for 1 min (annealing). The amplification steps were followed by a melt dissociation step to check for nonspecific products formation. Three replicates were performed for each sample and controls. Additionally, an analysis of the melting curve was performed, and qPCR products purity were controlled by 2% (*w/v*) agarose gel electrophoresis. Bacteria quantification was carried out using a standard calibration curve. Briefly, the DNA of each bacteria was obtained from a specific bacterial monoculture with a known colony-forming unit (CFU) per mL and used to create a standard calibration (Table 3). For each set of primers, five decimal dilutions of specific bacterial DNA were prepared to create the standard curve, which correlates the cycle threshold (Ct) values with the known log (CFU/mL).

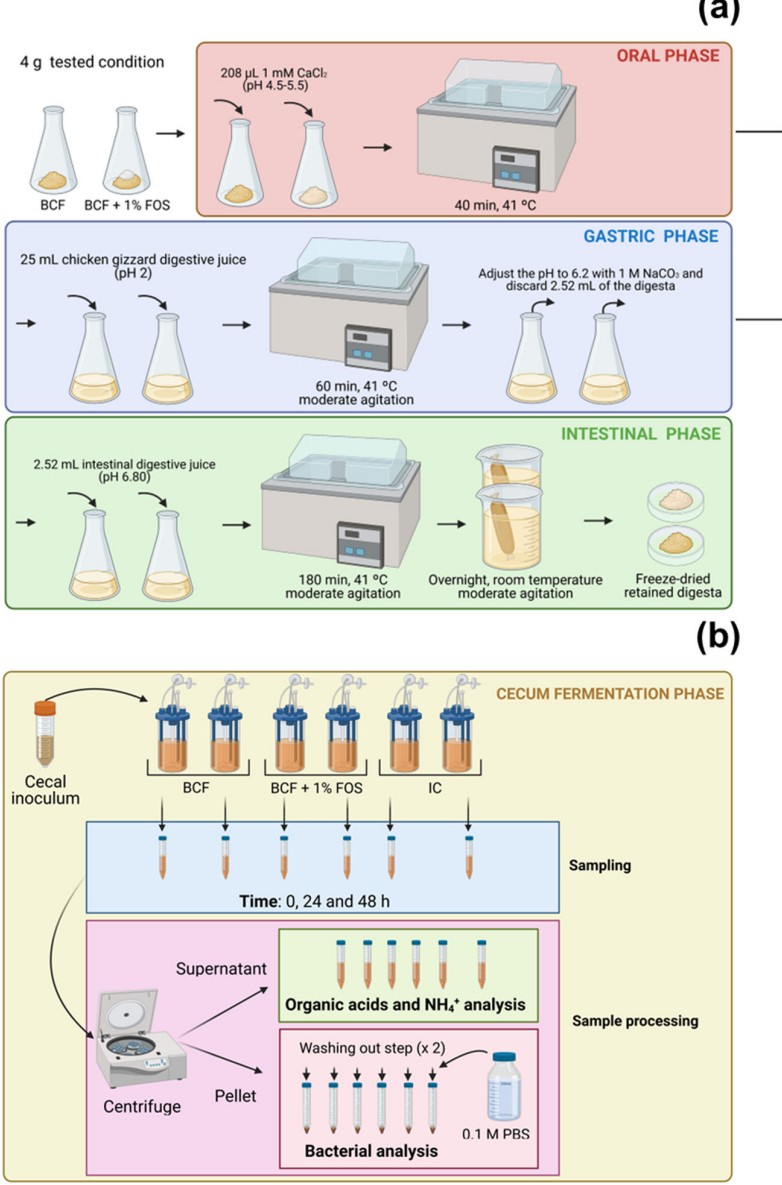

**Figure 3.** (**a**) Chicken GIT simulation protocol flowchart and (**b***) in vitro* fermentation experiments (cecum fermentation phase) and sample processing performed in this study.

### 2.4.4. Determination of Organic Acids Produced during *In Vitro* Fermentation

The supernatants collected after centrifugation were filtered (0.22 μm) and directly analyzed by HPLC in duplicates, as described by de Carvalho et al. (2022) [36]. Lactate, acetate, propionate, butyrate (2 mM to 80 mM), and DMSO (2 mM to 650 mM) were identified and quantified using their corresponding calibration curves. Organic acids and DMSO were expressed in mM.

### 2.4.5. Measurement of Total Ammonia Nitrogen Concentration

An ion-selective electrode 9663, at a constant temperature (room temperature, 20 °C) and pH (6.0–7.0), was used for the measurement of ammonium concentration according to [37]. For quantification, a standard calibration curve of $NH_4Cl$ (2 mM to 55 mM) was used. Total ammonia nitrogen concentration was calculated according to the equation below [38]:

$$\frac{\left[NH_4^+\right]}{\left[NH_3 + NH_4^+\right]} = 1 - \frac{1}{1 + 10^{pKa - pH}}$$

where $[NH_4^+]$ is the ammonium ion concentration, $[NH_3 + NH_4^+]$ is the total ammonia nitrogen concentration, and *pKa* is the acid dissociation constant that can be expressed as a function of temperature (*T*) using the following equation [38]:

$$pKa = 4 \times 10^{-8}T^3 + 9 \times 10^{-5}T^2 - 0.0356T + 10.072$$

### *2.5. Statistical Analysis*

Statistical analysis was carried out using IBM SPSS Statistics 27 software (IBM, Chicago, IL, USA). Data's normality was evaluated using Shapiro–Wilk's test. As all data followed a normal distribution, means were compared considering a 95% confidence interval, using one-way ANOVA coupled with Tukey's post-hoc test.

**Table 3.** Bacteria monocultures used as genomic DNA standard for calibration curves, their growth conditions and group-specific primers based on 16S rDNA sequences to profile cecal fermentation samples. F (Forward); MHB (Mueller–Hinton broth); MRSB (de Man, de Rogosa, and Sharpe broth); R (reverse); TSB (tryptic soy broth).

| Primer | Target Organisms | Genomic DNA Standard | Media Broth | Media Agar | Incubation Conditions | Sequence (5′-3′) | Amplicon Size (bp) | Reference |
|---|---|---|---|---|---|---|---|---|
| Firm | Firmicutes | *Lactobacillus gasseri* DSM 20077 | MRSB + 0.1% (*w/v*) cysteine | MRSA + 0.1% (*w/v*) cysteine | Anaerobic 37 °C 2 days | **F:** ATG TGGTTTAATTCGAAGCA **R:** AGCTGACGACAACCATGCAC | 126 | [39] |
| Lac | *Lactobacillus* group | | | | | **F:** CACCGCTACACATGGAG **R:** AGCAGTAGGGAATCTTCCA | 341 | [40,41] |
| Bdt | Bacteroidetes | *Bacteroides intestinalis* DSM 17393 | TSB + 5% (*v/v*) DSB | CBA + 5% (*v/v*) DSB | | **F:** CATGTGGTTTAATTCGATGAT **R:** AGCTGACGACAACCATGCAG | 126 | [39] |
| Bac | *Bacteroides* | | | | | **F:** GAAGGTCCCCCACATTG **R:** CGCKACTTGGCTGGTTCAG | 103 | [42] |
| Bif | *Bifidobacterium* | *Bifidobacterium animalis* ssp. *lactis* BB-12 DSM 15954 | MRSB + 0.1% (*w/v*) cysteine | MRSA + 0.1% (*w/v*) cysteine | | **F:** CGCGTCYGGTGTGAAAG **R:** CCCCACATCCAGCATCCA | 244 | [43] |
| Enb | *Enterobacteriaceae* family | *Salmonella enteritidis* subsp. *enterica* ATCC 13076 | MHB | MCA | | **F:** CATTGACGTTACCCGCAGAA-GAAGC **R:** CTCTACGAGACTCAAGCTTGC | 195 | [44] |

## 3. Results

### 3.1. Cecal Inoculum Viability

The viability of the cecal inoculum (fresh and frozen) was evaluated by culture dependents methods, using different culture media, according to the target group (Table 1). Figure 4 and Table S2 present the bacterial viable cell counts of inoculum A, inoculum B, and inoculum C. Inoculum A was the condition with the highest amount of viable bacterial cells in all culture media used ($p < 0.05$).

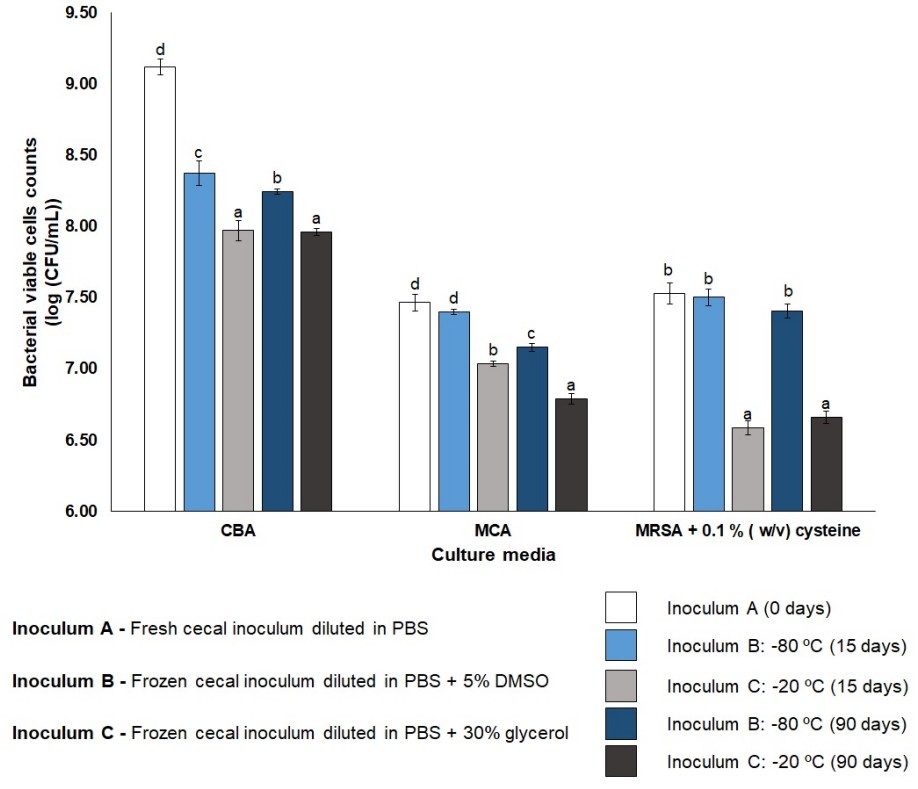

**Figure 4.** Bacterial viable cell counts (log (CFU/mL), mean ± SD) of fresh cecal inoculum (inoculum A) and frozen cecal inoculum (inoculum B and C) for 15 and 90 days. Different letters mark statistically significant ($p < 0.05$) differences between each condition when plated in each culture media.

For total anaerobic bacteria (CBA), the condition with the highest values was condition inoculum A (9.12 log (CFU/mL)), followed by inoculum B at 15 days (8.37 log (CFU/mL)), inoculum B at 90 days (8.24 log (CFU/mL)), and finally, the condition with the lowest values was inoculum C at 15 and 90 days (7.97 and 7.96 log (CFU/mL), respectively) ($p < 0.05$).

For Gram-negative/enteric bacteria (MCA), inoculum A (7.42 log (CFU/mL)) was not statistically different ($p > 0.05$) for inoculum B at 15 days (7.40 log (CFU/mL)); however, at 90 days, inoculum B (7.15 log (CFU/mL)) was statistically different ($p < 0.05$) for the other two conditions mentioned before. Inoculum C presented the lowest values of viable Gram-negative and enteric bacteria, with the condition stored at 15 days (7.04 log (CFU/mL)) being statistically different ($p < 0.05$) from that stored at 90 days (6.79 log (CFU/mL)).

For LAB/Bifidobacterium (MRSA + 0.1% ($w/v$) cysteine), inoculum B (stored at −80 °C), until 90 days, was not significantly ($p > 0.05$) different from the fresh inoculum, while inoculum C (stored at −20 °C) at 15 and 90 days had a lower amount of these bacteria compared with inoculum A and inoculum B ($p < 0.05$).

### 3.2. DMSO Wash-Out Confirmation

Inoculum B was submitted to two-DMSO wash-out cycles with 0.1 M PBS to reduce the presence of DMSO in the inoculum. The DMSO concentration present in the cecal inoculums

decreased significantly in each wash-out cycle (Table S3). After the first centrifuge (no washing step), the cecal inoculums had 596 mM of DMSO. In the first washing cycle, the supernatant of the cecal inoculum had 41 mM of DMSO, while the second wash cycle had 6 mM of DMSO. In the cecal fermentations with inoculum B, at time 0 h, the presence of DMSO was not detected, showing the efficacy of the washing method applied.

### 3.3. Chicken GIT Model Development

The duplicates of each feed condition that were submitted to the oral, gastric, and intestinal phases showed a similar appearance between themselves at the end of these phases. The protocol for these mentioned phases took four to five days to accomplish. Each duplicate started with 4 g of feed and finished the gastrointestinal digestion and absorption part (after the freeze-dryer) of the GIT model developed with approximately 3 g. Each feed condition's duplicates were manually homogenized and properly stored at room temperature in a dry condition before their use in the cecum fermentation phase, which takes another four to five days to finish. Therefore, the total developed protocol, accounting for all phases, takes eight to ten workdays to be performed.

### 3.3.1. Bacterial Profile of the Cecum Fermentations

Figure 5a–f and Table S4 show the bacterial concentration of the different bacterial populations during the cecum fermentations of the three conditions tested (1-SCF, 2-SCF + FOS, and 3-IC) with two inoculums, inoculum A (fresh) and inoculum B (frozen).

At 0 h, the primers that quantified the largest DNA amount were Firmicutes (Firm, average of 7.6 log (CFU/mL)) followed by *Enterobacteriaceae* family (Enb, average of 6.9 log (CFU/mL)). Bacteroidetes (Bdt), *Lactobacillus* group (Lac), and *Bacteroides* (Bac) had the same average bacterial concentration (6.6 log (CFU/mL)). *Bifidobacterium* (Bif) was the bacterial group studied with the lowest quantity (average of 2.5 log (CFU/mL)) present in the cecal fermentations at time 0 h. Significant differences ($p < 0.05$) were found at time 0 h, between the cecal fermentation that used inoculum A and inoculum B for all bacterial populations assessed in this study, except for Bif ($p > 0.05$).

The IC condition stood out the most, because either with inoculum A or B, the cecal fermentations registered the lowest values of quantified Lac (Figure 5c) and Bif (Figure 5e), during the full course of the fermentation process (i.e., 0 to 48 h).

Concerning the initial concentration present of Firm (Figure 5a), for time 24 h, the bacterial concentration decreased in all conditions, except for IC with inoculum B which maintains the bacterial concentration. The condition with the highest amount ($p < 0.05$) was IC with inoculum B, followed by IC with inoculum A, while the other conditions (i.e., SCF and SCF + FOS with inoculum A or B had lower values of Firm and were not significantly different ($p > 0.05$) between each other. At time 48 h, the conditions with the highest value of Firm ($p < 0.05$) were SCF with inoculum A and B, followed by both conditions of SCF + FOS and the condition with lowest values were both IC. The condition SCF, for both inoculums, is the only condition at time 48 h that does not have a reduction in Firm in regard to time 0 h.

In Figure 5b, related to the quantification of Bdt, at time 24 h, the condition IC with inoculum A and B was the condition with the lowest amount quantified ($p < 0.05$), SCF with inoculum A was the condition with the highest presence of Bdt ($p < 0.05$), while the remaining conditions (i.e., SCF with inoculum B and SCF + FOS with inoculum A and B) were not statistically different ($p > 0.05$). At 48 h, none of the conditions were significantly ($p > 0.05$) different from each other with both inoculums.

Related to quantifications for the concentration of Lac in cecal fermentations performed (Figure 5c), at time 24 h, there are significant differences between the inoculums used in the three conditions assessed and not between conditions with the same inoculum. The conditions (i.e., IC, SCF, and SCF + FOS) inoculated with inoculum A had a smaller presence of Lac than the same conditions that were inoculated with inoculum B ($p < 0.05$). At times 24 and 48 h, the IC with both inoculums was the condition with a lower amount of Lac

present ($p < 0.05$) while SCF and SCF + FOS were not significantly different from each other at these fermentation time points.

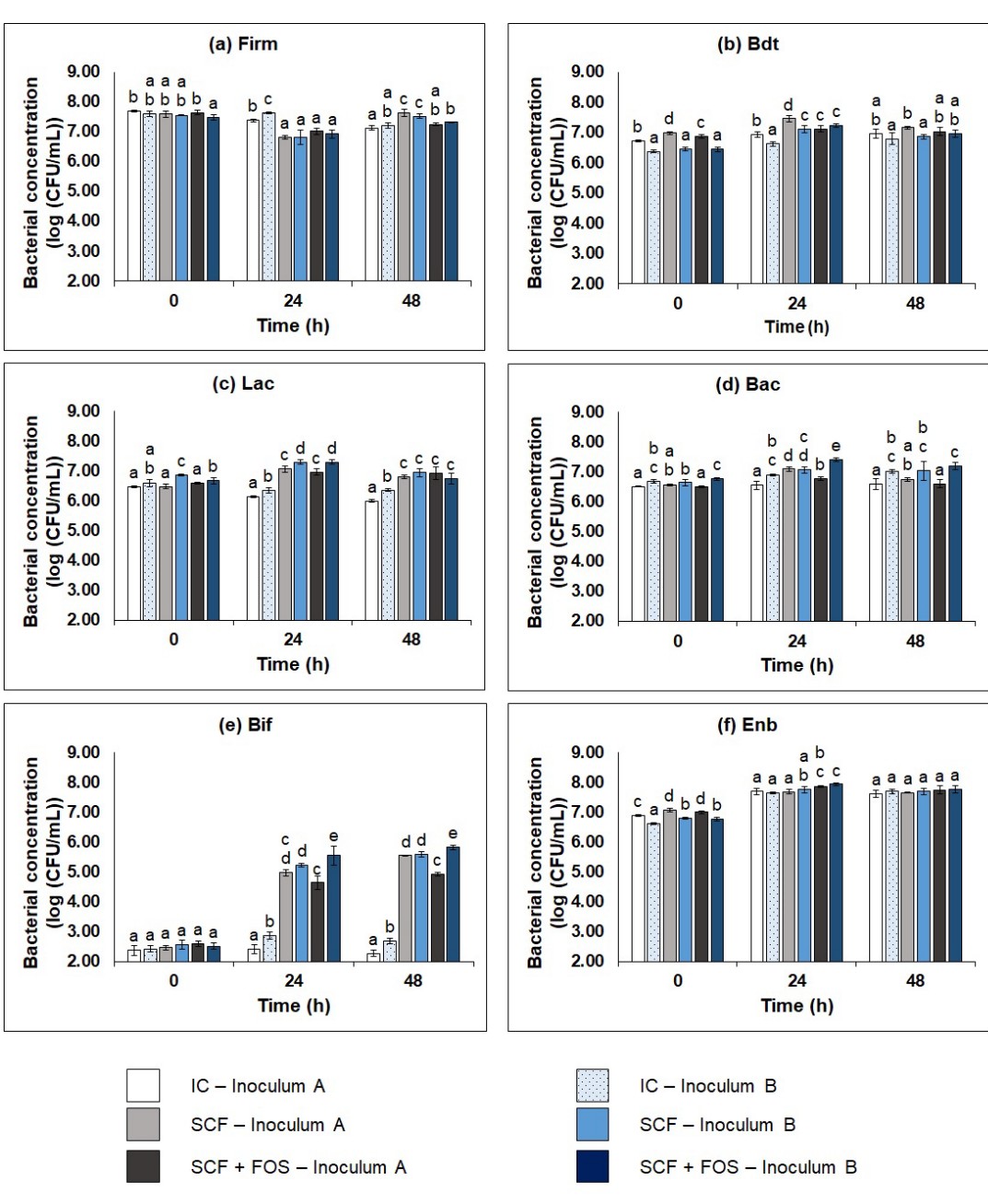

**Figure 5.** Bacterial quantification (log (CFU/mL), mean ± SD) of the different bacterial populations present in the cecal fermentation for the different conditions performed: (**a**) Firm, (**b**) Bdt, (**c**) Lac, (**d**) Bac, (**e**) Bif, and (**f**) Enb. Different letters mark statically significant ($p < 0.05$) differences between each condition at each primer.

The quantification of Bac In the cecal fermentation (Figure 5d) presents significant differences for the same condition in the different inoculums, as well as between conditions with the same inoculum at 24 h. The only two conditions with a significant difference ($p < 0.05$) were the SCF + FOS with inoculum B, which had the highest concentration of Bac at 24 h (7.41 log (CFU/mL)), while IC with inoculum A was the condition showed the lowest values (6.55 log (CFU/mL)). At 48 h, the only significant difference ($p < 0.05$) recorded is related to the inoculum used (inoculum B had higher growth of Bac) since

the different conditions tested with the same inoculum do not have significant differences ($p > 0.05$).

Regarding the quantification of Bif (Figure 5e), at 24 and 48 h, the conditions with inoculum B had higher bacterial growth ($p < 0.05$) than with inoculum A, except SCF + FOS with inoculum B. The conditions with the lowest amount of Bif, at 24 and 48 h, were the IC for both inoculums ($p < 0.05$). At 24 h, SCF + FOS had a higher concentration ($p < 0.05$) of Bif than SCF with inoculum B, while with inoculum A these two conditions were not statistically different ($p > 0.05$). At 48 h, the concentration of Bif in SCF + FOS was higher ($p < 0.05$) than SCF with inoculum B, while in inoculum A, SCF + FOS had a lower concentration ($p < 0.05$) of Bif than SCF.

Related to the Enb quantification during the experiment, the only significant differences ($p < 0.05$) were verified at 24 h at the condition SCF + FOS for both inoculums that had higher growth in the same inoculum than IC and SCF. However, at 48 h, all conditions were tested and neither inoculum was statistically different ($p > 0.05$) from each other.

### 3.3.2. Metabolic Profile of Cecum Fermentations

Figure 6a–d and Table S5 show the concentration of lactate, acetate, propionate, and butyrate produced throughout the 48 h of cecum fermentation. The organic acids produced in higher amount during the fermentation was lactate, followed by acetate, propionate, and butyrate. The conditions with the highest production of SCFA (i.e., acetate + propionate + butyrate) at the end of the fermentation for inoculum A and B were SCF + FOS (75 and 78 mM, respectively) > SCF (71 and 71 mM, respectively) > IC (43 and 32 mM, respectively).

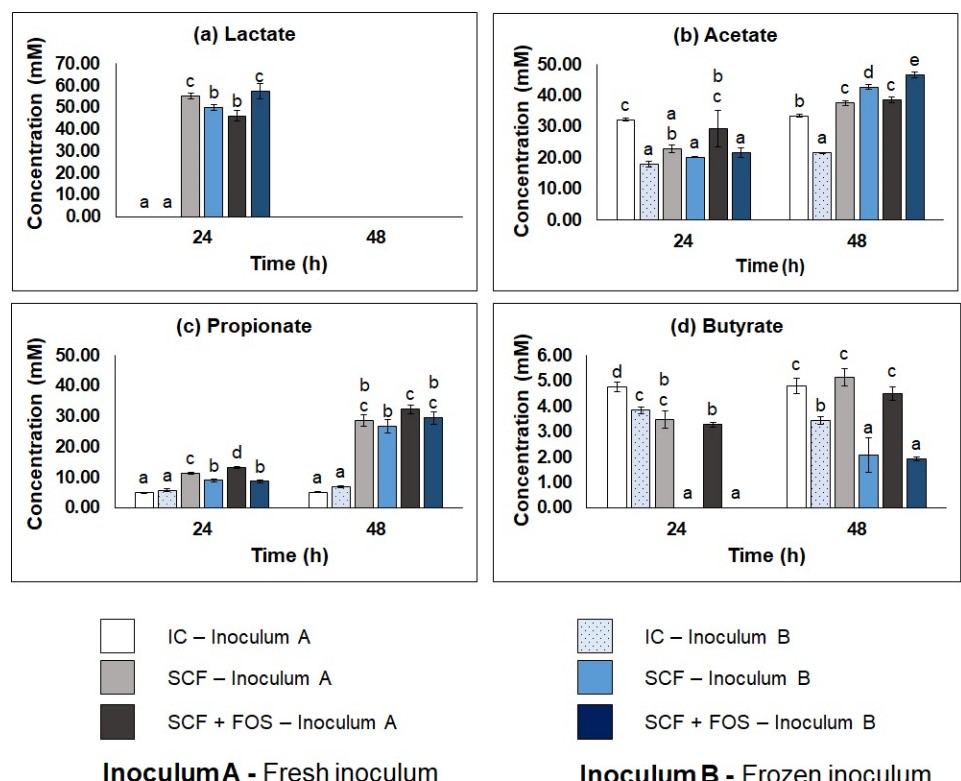

**Figure 6.** Concentrations (mM, mean ± SD) of the different organic acids produced during the cecal fermentation for the different conditions performed: (**a**) Lactate; (**b**) acetate; (**c**) propionate; (**d**) butyrate. Different letters mark statically significant ($p < 0.05$) differences between each condition at each sampling time.

Lactate (Figure 6a) had a different production pattern from the other organic acids assessed, with production (i.e., lactate detected at 24 h) and consumption (i.e., lactate not

detected at 48 h) during the experiment, except for the IC condition, in which the production of lactate was not detected with both inoculums. Meanwhile, the production of the different SCFA increased over time. Fermentation with inoculum A, SCF was the condition with the highest production of lactate ($p < 0.05$), while fermentation with inoculum B, SCF + FOS was the condition that had more lactate produced ($p < 0.05$).

Acetate production (Figure 6b) at 24 h was higher ($p < 0.05$) with inoculum A than with inoculum B, but at 48 h acetate production was higher ($p < 0.05$) with inoculum B instead of inoculum A, except for the IC condition that verified higher production of acetate with inoculum A. At 24 h, the condition with the highest production of acetate ($p < 0.05$) with inoculum A was IC, while in inoculum B there were no differences ($p > 0.05$) between the conditions tested. At 48 h, with inoculum A, the only statistical difference verified was between the IC condition with the two remaining conditions, which had a lower acetate production ($p < 0.05$), while in inoculum B, the SCF + FOS condition had the highest production followed by SCF and the IC returned to be the condition with the lowest acetate production ($p < 0.05$).

Regarding propionate production (Figure 6c), a higher production was observed with inoculum A than with inoculum B ($p < 0.05$), except for the IC condition, where there were no differences ($p > 0.05$) between the two inoculums. With inoculum A, at 24 h, SCF + FOS had a higher concentration of propionate produced than SCF however with inoculum B there is no difference between these two conditions. At 48 h, the same tested condition with the two inoculums did not have different production of propionate and the only difference between the condition was the production of propionate in the IC condition was lower ($p < 0.05$) than SCF and SCF + FOS.

Butyrate production obtained the most dissimilar results of using inoculum A and B (Figure 6d). By using these two inoculums, it was observed that at times 24 and 48 h inoculum A had a greater production of butyrate ($p < 0.05$), and in inoculum B, for SCF and SCF + FOS conditions, butyrate was only present at 48 h. The IC condition in both inoculums was the condition that had a higher concentration of butyrate produced at time 24 h. However, at time 48 h, the butyrate production in IC condition was not different ($p > 0.05$) for SCF and SCF + FOS with inoculum A, while with inoculum B the IC condition was once again the condition with the highest concentration of butyrate. Using the same inoculum, in all time points, SCF and SCF + FOS did not have different ($p > 0.05$) butyrate production.

### 3.3.3. Total Ammonia Nitrogen Profile of Cecum Fermentations

Figure 7 and Table S6 display the concentration of total ammonia nitrogen produced during the cecal fermentations. At the beginning of fermentation, at time 0 h, there were differences ($p < 0.05$) in concentrations present between the same condition with inoculum A and inoculum B; however, between conditions with the same inoculum used there was no difference ($p > 0.05$) in the concentration of total ammonia nitrogen. The production of total ammonia nitrogen increased throughout the experiment, observing that the highest production value in each condition in both inoculums was at 48 h.

At 24 and 48 h, fermentations carried out with inoculum A produced more total ammonia nitrogen than with inoculum B in the same condition ($p < 0.05$). Between conditions in both inoculums, the IC condition produced a higher amount of total ammonia nitrogen than the other two conditions in both time points ($p < 0.05$). With inoculum A, SCF and SCF + FOS were only statistically different at 48 h in which SCF + FOS was the condition with the lowest production of total ammonia nitrogen; however, with inoculum B, SCF and SCF + FOS at any point of the experiment had different productions of SCF and SCF + FOS ($p > 0.05$).

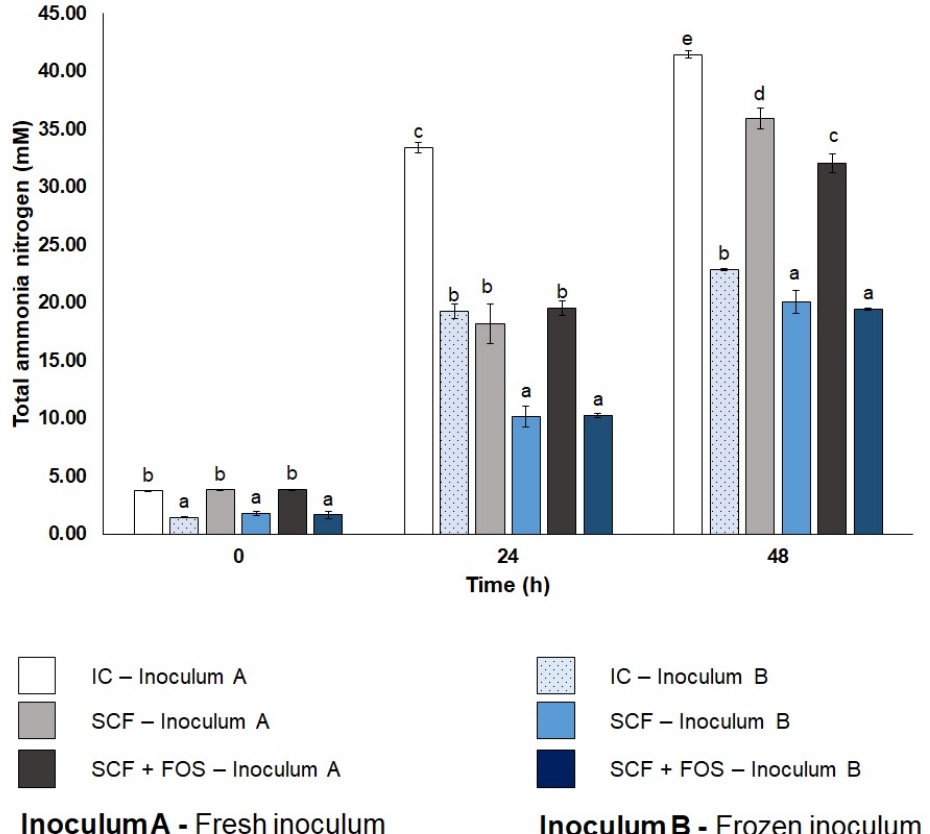

**Figure 7.** Concentration (mM, means ± SD) of total ammonia nitrogen produced during cecal fermentation with fresh (A) and frozen (B) inoculum. Different letters mark statistically significant ($p < 0.05$) differences between each condition at the same sampling time.

## 4. Discussion

This research work focused on three major goals: (1) the development of the chicken GIT model to be used as a screening method to assess the impact of feed additives supplementation in diets. The choice of FOS relies on the fact that this is a commonly used prebiotic in poultry nutrition and has been proven to have capacity to promote beneficial intestinal bacteria (e.g., Lac and Bif) on cecum microbiota and increase the production of the SCFA and lactate which decreases the intestinal pH inhibiting the growth of pathogenic microorganisms (e.g., *Salmonella* and *Escherichia coli*), among other activities (e.g., vitamin production) that will impact positively the animal intestinal physiology [17,45]. The second goal was (2) the evaluation of two different preservation methods of the cecal inoculum to be used in the cecal fermentation stage. This evaluation was performed, in search of a recommended preservation method to be applied on the chicken cecum inoculum, allowing more practicability on the developed GIT model, since a "ready-to-use" inoculum would be available. The third goal was (3) the comparison between fresh and frozen inoculum (the best preserved one) fermentation profiles, related to bacteria and metabolite production (i.e., organic acids and ammonia). This study was done to better understand the influence of the inoculum proprieties and conditions on results when applied to the developed GIT model.

### 4.1. Chicken Inoculum Preservation Method Impact on Bacteria Viability

To be able to closely simulate the cecum fermentation in broiler chickens, cecal contents from broiler chickens were collected to prepare an inoculum that represents the cecum microbiota present in these animals. Due to the small amount of cecal content that is collected from each animal cecum (average ~5 g, as seen in Table S1), a pooled cecal

inoculum was prepared, allowing for an increase in the availability of inoculums to be used for several sets of experiments.

The cecum content collected were all from Ross 308 broiler chickens, within the same age interval and supplier (same diet), as it is the most common breed for human consumption in this region and thus is in higher availability in the slaughterhouse. The use of the animal's cecal content instead of their fecal material is more adequate to simulate the physiological conditions of cecal fermentation since there is a difference between the bacterial communities in these biological samples [35]. The preparation of a pool inoculums for the GIT models application and utilization of one single homogenous inoculum, provides more reliable and reproducible results from these models as reduce inter-donor variability and minimize the interference of outlier's donors [27].

In this study, the cecal contents were diluted at 10% (*w/w*) to make the cecal inoculum to be used in the cecum fermentations. The cecal inoculum A (fresh cecal inoculum) contained 9.12 log (CFU/mL) of total anaerobic bacteria, 7.47 log (CFU/mL) of Gram-negative and enteric bacteria, and 7.53 log (CFU/mL) of LAB/*Bifidobacterium*. This means that the cecal content used contained 10 log CFU/mL, 8–9 log CFU/mL of Gram-negative/enteric bacteria and LAB/*Bifidobacterium*. Also, the bacterial viability of inoculum B (PBS + 5% DMSO) stored at −80 °C and inoculum C (PBS + 30% glycerol) stored at −20 °C were verified at 15 and 90 days of storage. The results indicate that storage at −80 °C with 5% DMSO is the best option compared to storage at −20 °C with 30% glycerol. For inoculum B, it was observed that the only bacterial group that did not decrease ($p > 0.05$) over the storage time was LAB/*Bifidobacterium*, while the remaining bacterial groups showed a reduction ($p < 0.05$) during the storage in relation to inoculum A. This indicates that freezing cecal inoculum in PBS + 5% DMSO at −80 °C for at least 90 days does not affect LAB/*Bifidobacterium* viability.

The evaluation of different bacteria viability present in the cecum inoculums, through the utilization of culture-dependent methods, is a recommended and common practice in the clinical, veterinary, and food safety fields, however, it has their own limitations, namely the inability of quantifying uncultured bacteria, which represent a significant portion of the overall bacterial population (i.e., ~60–80%) present in biological samples such as cecal material [46–48]. For this reason, platting was the chosen technique to measure the amount of culturable total anaerobic bacteria present in the inoculum, in addition to determining the amount of culturable Gram-negative/enterobacteria and LAB/*Bifidobacterium*, which are relevant bacterial groups for poultry health and their cecal fermentation process [49].

The results obtained from the bacterial viability of inoculum A (Figure 4 and Table S2) are within what would be expected for the cecum content of chickens between 32 to 42 days old (independently of their diet) according to the scientific literature [21,50]. In these studies, the total viable bacteria present are within a range of 10–11 log (CFU/mL), Gram-negative bacteria (coliforms) are between 7–9 log (CFU/mL), and LAB/Bifidobacterium are between 8–10 log (CFU/mL) [21,50].

Bacteria such as LAB/*Bifidobacterium* are vital when it comes to characterizing the intestinal microbiota responsible for the production of lactate and SCFA, important metabolites in the evaluation of the feed additive's impact on poultry health [20,51]. Therefore, maintaining the viability of these groups is crucial to replicate the metabolic processes (e.g., breakdown of carbohydrates and proteins, production of organic acids and ammonia) that occur in the cecum microbiota. Assessing its modulation in response to a studied feed condition validates the results that are dependent on the inoculum microbial activity.

The pooled cecum content collected was diluted in PBS to help homogenize the cecal inoculum and keep the inoculum pH stable (pH ~ 7). However, the use of saline solutions and storage at low temperatures can cause damage to bacteria cells, due to lesions on the bacteria's surface and the formation of intracellular ice particles [27]. Thus, when storing the inoculums, a cryoprotectant was added to the PBS solution to protect the microorganism's cell membranes during freezing.

The chosen cryoprotectant for the frozen inoculum (inoculum B) was DMSO at 5%, since it is one of the more effective intracellular cryoprotectants (even more than glycerol, a well-known and common intracellular cryoprotectant used in molecular biology) to diffuse through the cell membrane, changing the properties of the fluid and increase membrane glass-phase transition temperature resulting in the reduction of intracellular ice crystals [52,53].

Inoculum B was stored at −80 °C, as this temperature is the most suitable for long-term storage of biological samples like feces, cecum, or even ruminal content [24,54,55]. In previous works, the authors have managed to work with human fecal inoculum preserved at −20 °C in PBS + 30% glycerol for *in vitro* fermentation for up to 90 days [27,36]. In this study, the difference between the method of preservation of the authors (i.e., PBS + 30% glycerol at −20 °C) and the current one performed (i.e., PBS + 5% DMSO at −80 °C) was assessed at 15 and 90 days of storage, in which it was observed that the condition that best preserved all quantified bacteria (i.e., total anaerobic bacteria, Gram-negative/enteric bacteria, and LAB/Bifidobacterium) present in the pooled cecal inoculum was the one preserved with PBS + 5% DMSO at −80 °C (Figure 4 and Table S2).

These results for inoculum bacterial viability confirm that up to 90 days of the preservation of cecal inoculum at −80 °C in PBS + 5% DMSO is the most suitable approach for preserving chicken cecal inoculum for *in vitro* fermentations in comparison with the previous preservation method performed by the authors in the human fecal inoculums.

### 4.2. Gastrointestinal Model and Cecum Microbiota Assessment

After thawing the stored frozen inoculums (Inoculum B), a DMSO washing protocol (two-DMSO washing-out cycles) was carried out with the purpose of removing the presence of DMSO in the inoculum meant for cecal fermentations (at 41 °C). This washing protocol was successful to the point that DMSO was not detected or quantifiable by the HPLC equipment for the reactors that carried out cecum fermentations with inoculum B.

Although the condition with DMSO was the most adequate preservation method tested in this study, the use of DMSO should be carefully managed, since DMSO is toxic for the bacterial cells at a certain percentage (it is common to be used between 5 to 10% as a cryoprotective agent in a wide range of biological samples) and at a temperature above 4 °C degrees. For that reason, this DMSO washing protocol was done [56,57].

The development of the gastrointestinal simulation model took into account the events that the ingested feed undergoes along the chicken GIT. To test the practicality and reproducibility of the model, two types of feed (i.e., SCF and SCF + FOS) were submitted to the GIT model developed, as shown in Figure 3. The duplicates of the digested samples for each condition were identical between itself showing that the digestive protocol was reproducible. The purpose of assessing SCF was to understand if it can be used as a standard response to a specific feed matrix. SCF + FOS was used as positive control and IC was used as negative control.

In this study, for the cecum fermentation phase, it was proposed to use four reactors for each condition (two reactors with fresh inoculum and two reactors with frozen inoculum), by doing duplicates of this protocol (4 g of feed matrix for each condition), were enough to have a quantity to be used on this number of reactors, being simple, easy to perform, reproducible, having a fast protocol to perform, and being inexpensive in regards to animal trials.

For the bacterial quantification during cecum fermentation, the qPCR technique was used because it is a suitable technique for quantifying phyla and genus. The selected bacterial groups (Table 3) were chosen due to these microorganisms being common members of the cecal microbiota and playing important functions in the host's nutrition (e.g., nutrient digestion/absorption), physiology (e.g., growth performance, and intestinal morphology), and immune system (e.g., innate and acquired immune response), thus influencing the host's health status [45,58].

In this study, at time 0 h (Figure 5 and Table S4), Firm was more represented in the cecal microbiota than Bdt (at time 0 h: Firm—7.6 log (CFU/mL) and Bdt—6.6 log (CFU/mL)), with a high predominance of Lac, Bac, and members of Enb in the cecum microbiota, and a lower presence of Bif in regard to mentioned assessed bacterial groups. Also, the results obtained for Lac and Enb correspond to the results obtained from the inoculum viability study, in which Gram-negative/enteric bacteria (Enb are also Gram-negative/enteric bacteria) and LAB/Bifidobacterium (Lac are LAB) had similar bacterial viable cells counts between each other.

These results related to chicken cecum microbiota are in accordance with the scientific literature, in which the most predominant phylum in the chicken cecum microbiota is Firm, representing 70%, followed by Bdt, representing 12.3%, with a high predominance of genus that are quantified by Lac, Bac, and Enb [25,50]. The low presence of Bif in this study can be related to the absence of Bif in the early ages of the chicken cecum, only appearing at four weeks old, mostly due to environmental factors, dietary practices, and breed [25,50]. Therefore, at time 0 h, the results are aligned with what would be expected for chicken's cecum microbiota at six weeks of age.

In regards to the bacterial quantification during the cecum fermentation (Figure 5 and Table S4), it was observed that the use of inoculum A and B had identical bacterial behaviors for the same studied conditions (i.e., IC, SCF, and SCF + FOS), except for Bac and Bif. Between SCF and SCF + FOS conditions, the only difference ($p < 0.05$) observed, for both inoculums, was related to the growth of Enb at 24 h and Firm at 48 h. This study verified, that up to 48 h, there is still bacterial growth happening throughout fermentation, and the IC condition was the only condition that observed a very small promotion, or the absence of promotion, in some bacteria populations (especially, Lac and Bif) for both inoculums. This result related to the IC condition is due to the absence of the addition of nutrients that would enable the maintenance or growth of the cecal bacteria. Regardless of the inoculum used (A or B), it was observed that the condition with the highest promotion of Bac at 24 h had the highest growth of Bif at 48 h (i.e., for inoculum A was SCF while for inoculum B was SCF + FOS). The addition of FOS, for inoculum B, promoted the growth of Bif. Also, SCF + FOS, for both inoculums, had a lower amount of Firm than SCF.

The bacteria that belong to Bac and Bif have in common saccharolytic activity and are directly related to the production of SCFA during carbohydrates fermentation, especially propionate (Bac) and acetate (Bac and Bif) [59,60]. According to several authors [10,45,61], dietary FOS supplementation increases Lac and Bif diversity, while reducing *Escherichia coli* (a genus that belongs to Enb) diversity in the chicken gut; however, in this study, only the promotion of Bif was observed (only for inoculum B). Also in this study, the addition of FOS verified the reduction of Firm, which are bacteria positively related to the degradation of polysaccharides and associated with the extraction of energy/calories of feed (more efficient than Bdt), animal weight gain, and/or obesity [10,62–64].

Bacterial growth during cecum fermentations promotes the production of metabolites such as SCFA and ammonia, through the degradation of fiber, protein, and peptides. Acetate, propionate, and butyrate account for 90–95% of the total SCFA produced. Their beneficial effects range from energy generation to regulation of intestinal blood flow, mucin production, proliferation and enterocyte growth, and intestinal immune response, among others [20,25,51]. Other important metabolites are also produced by the cecum microbiota, namely lactate, an organic acid that is directly involved in the production of SCFA (i.e., acetate, propionate, and butyrate) and is produced by bacteria such as Lac and Bif [17,65].

In Figure 6a and Table S5, for SCF and SCF + FOS, with both inoculums, a cross-feeding phenomenon occurred, in which lactate was produced (by e.g., Lac and Bif) until 24 h, but after 48 h, it disappeared due to its conversion (by e.g., Firm and Bac) into acetate, propionate, or butyrate, as verified by the increase in these acids at 48 h. This process is important because it limits the accumulation of lactate, allowing it to prevent metabolic acidosis [65]. The only condition in which there was no lactate production was the IC for both inoculums, and this result makes sense since it was the only condition in which

there was no growth promotion of Lac and Bif, which are the main lactate producers in the cecum microbiota [51,65]. It is also observed that the conditions with the highest lactate concentration produced at 24 h in their respective inoculums (i.e., SCF for inoculum A and SCF + FOS for inoculum B) correspond to the conditions with the highest growth of Bac at 24 h and Bif at 48 h. Therefore, this production of lactate is directly related to the growth of Bif, which had an influence on the growth of Bac.

The production of SCFA was different between inoculum A and B, for the same condition except for propionate. There is a significantly ($p < 0.05$) lower production of propionate and butyrate at 24 h with inoculum B (Figure 6c,d and Table S5) which affects the final concentration at 48 h of butyrate ($p < 0.05$) but not of propionate ($p > 0.05$). It appears that the use of frozen inoculums in the first hours of fermentation has lower production of SCFA, but during the experiment, it is equivalent to the production of fresh inoculums, except for the production of butyrate, showing that butyrate producers are negatively affected by the storage of the inoculum at $-80\ °C$ for up 90 days. In Table S5, for the sum SCFA and ratio acetate: propionate: butyrate (A: P: B), the condition that has the highest production of the SCFA was the SCF + FOS condition, with both inoculums, detecting that the addition of FOS at 1% in SCF increases the production of SCFA (specifically, acetate). Also, in Table S5, it was observed that the condition with feed with inoculum B had values of the ratio A: P: B that were very different from the other conditions, which is related to the low production of butyrate in these conditions throughout the fermentation.

The acetate production behavior observed in the results may be related to the bigger growth promotion of Bac and Bif in this condition with inoculum B, because these bacteria are associated with acetate production [59,60]. The ratio A: P: B (Table S5) for the IC condition is within the values found in a previous study related to chicken cecum microbiota (3:1:1 to 8:2:1) [66].

In Figure 7 and Table S6, it is observed that the condition with the highest production ($p < 0.05$) of ammonia was the IC condition. The addition of FOS in SCF, in inoculum A, had a lower ($p < 0.05$) ammonia production than SCF; however, this effect was not statistically relevant ($p > 0.05$) for inoculum B. Also, in Figure 7 and Table S6, inoculum B had lower production of ammonia ($p < 0.05$), for the same condition, than inoculum A, meaning that the storage of inoculum negatively affected the proteolytic activity of the cecal inoculum bacteria.

This result of ammonia production can be explained by the results obtained for SCFA production (Figure 6 and Table S5). During the fermentation, the production of SCFA causes a reduction in the pH values which reduces the formation of ammonia from the amino acid deamination [26,67]. The IC was the condition with the lowest value of SCFA production; therefore, these results could explain the reason why the other two conditions had lower ammonia production.

Ammonia is a metabolite, which, depending on its concentration in the gut, can be toxic to the host, produced by proteolytic bacteria present in the intestinal microbiota that use ammonia as a nitrogen source and it is excreted through the stool and urine or absorbed in the gut [26,67]. Other harmful effects of the intestinal production of ammonia in the poultry industry are related to the negative impact ammonia has on the environment and the health/welfare of animals and poultry house workers [68,69].

The fermentation profile obtained from inoculums A and B, even though from the same biological samples (i.e., the pool of cecal contents from broiler chickens), had different bacterial growth patterns and metabolic activity during the *in vitro* fermentation experiments for the same conditions assessed (i.e., SCF, SCF + FOS, and IC), showing that the use of the same inoculum with different storage conditions (i.e., fresh and storage at $-80\ °C$ for 90 days) can impact the outcome of the assay. The results obtained in this work demonstrate that this chicken GIT developed has, as one of its limitations, the inoculum that is used in cecal fermentations, which is dependent on its preparation and storage, directly influencing the outcome obtained. Therefore, the results obtained from inoculums

at different stages of preservation should not be compared since the microbiota present in these inoculums have different behaviors, as seen in this study.

Regardless of the result obtained in these experiments, with any of the inoculum used, it should be noted that these results were obtained using an *in vitro* methodology and therefore must be further validated by *in vivo* trials, as these will ultimately test the beneficial effects of feed additive supplementation [46].

The chicken GIT model developed, optimized, and implemented by the authors has great potential as a tool to evaluate chicken feed impact on cecal microbiota, enabling a rapid preliminary screening of novel ingredients potential benefits. Thus, bringing valuable insights to poultry nutrition and microbiology research. However, *in vitro* studies do have its limitations when it comes to represent live systems; therefore, it is recommended that *in vivo* studies are carried out simultaneously, for comparison. *In vivo* studies will enable a better understanding of the "real life" context and help to validate the *in vitro* assessments, as a tool for preliminary studies, avoiding the costs, time, and animal welfare concerns that *in vivo* experiments apport.

### 5. Conclusions

The developed GIT *in vitro* model was shown to be a reliable preliminary assay for the intended purposes, as it represents the complete GIT system that closely mimics the live processes of the chicken GIT. This model can have different applications, but overall, it can provide valuable information for novel and well-established feed ingredients used in chicken diets, namely their potential in regards to intestinal microbiota modulation and ultimately animal performance and health. This is a simple protocol to follow, reproducible, which uses low quantities of feed samples (i.e., 4 g), is not time consuming (i.e., 8–10 workdays to perform), and avoids excessive and unnecessary number of *in vivo* experiments.

This study also concludes that the preservation method proposed (diluting the cecal content in PBS + 5% DMSO and storage at −80 °C), is appropriate to maintain the inoculum viability up to 90 days, as the LAB/*Bifidobacterium* viability was unaltered when compared to the fresh inoculum. However, a small significant decrease in total bacteria and Gram-negative/enteric bacteria occurred.

Different fermentation profiles were obtained when using fresh and frozen inoculum from the same biological samples during the *in vitro* fermentation (i.e., cecum fermentation phase). Even though the same feed conditions were used, results obtained from inoculum with different storage conditions should not be compared since each microbiota present in the inoculum has its own metabolism and interaction patterns. As seen in this study, frozen inoculum had a lower production of butyrate and ammonia than fresh inoculum for the same feed conditions.

The methodological approach proposed in this study by the authors shows itself as a cost-effective preliminary screening, prior to *in vivo* studies, enabling the impact of ingredient supplementation in animal diets on the host's microbiota modulation and subsequent effects on their health and performance to be evaluated. This approach can pinpoint different outcomes and potential benefits of different feed supplementations. However, this methodology does not substitute the need to perform *in vivo* testing, as for any *in vitro* model it does not replicate all the biological live processes and interactions that occur *in vivo*.

**Supplementary Materials:** The following supporting information can be downloaded at: https://www.mdpi.com/article/10.3390/applmicrobiol3030066/s1, Table S1: Information regarding the selected broilers for the preparation of the cecal inoculum used in this study; Table S2: Bacterial viable cell counts (log (CFU/mL), mean ± SD) of the cecal inoculums in different culture media; Table S3: DMSO concentration (mM, mean ± SD) in the frozen cecal inoculum during the two washing-out cycles; Table S4: Bacterial quantification (log (CFU/mL), means ± SD) of the different bacterial populations in cecal fermentation for different conditions with fresh (A) and frozen (B) inoculum; Table S5: Concentration (mM, means ± SD) of the different organic acids

produced during cecal fermentation with fresh (A) and frozen (B) inoculum; Table S6: Concentration (mM, means ± SD) of total ammonia nitrogen produced during cecal fermentation with fresh (A) and frozen (B) inoculum.

**Author Contributions:** Conceptualization, N.M.d.C., D.L.O. and A.R.M.; methodology, N.M.d.C., C.M.C., C.C., M.A.D.S. and D.L.O.; validation, A.R.M.; formal analysis, N.M.d.C.; investigation, N.M.d.C., C.M.C., C.C., M.A.D.S. and D.L.O.; resources, M.E.P. and A.R.M.; data curation, N.M.d.C.; writing—original draft, N.M.d.C.; writing—review and editing, D.L.O. and A.R.M.; visualization, N.M.d.C.; supervision, D.L.O. and A.R.M.; project administration, M.E.P. and A.R.M.; funding acquisition, M.E.P. and A.R.M. All authors have read and agreed to the published version of the manuscript.

**Funding:** This research was funded by Fundo Europeu de Desenvolvimento Regional (FEDER), through the Programa Operacional Competitividade e Internacionalização (POCI) under the project Alchemy: Capturing High Value from Industrial Fermentation BioProducts (POCI-01-0247-FEDER-027578). The authors would also like to thank the scientific collaboration under the Fundação para a Ciência e Tecnologia (FCT) project UID/Multi/50016/2019.

**Institutional Review Board Statement:** This is an *in vitro* study which only used biological samples of animals slaughtered for industrial purposes under legal, appropriate, and safe conditions.

**Data Availability Statement:** The data presented in this study are available in the article or Supplementary Materials.

**Acknowledgments:** The authors would like to acknowledge Diana Valente, Inês Ribeiro and João Azevedo-Silva for all the assistance and support in optimizing the DNA extraction and quantification techniques and qPCR protocols.

**Conflicts of Interest:** The authors declare no conflict of interest.

## Appendix A

*Appendix A.1. Reagents/Chemicals Used in This Study*

- Acetic acid glacial (Sigma, St. Louis, MO, USA);
- Agarose—electrophoresis grade (Nzytech, Lisbon, Portugal);
- Ammonium chloride—$NH_4Cl$ (Mettler Toledo, Urdorf, Switzerland);
- Bile porcine- B8631 (Sigma, St. Louis, MO, USA);
- Bile salts (Sigma, St. Louis, MO, USA);
- Butyric acid (Sigma, St. Louis, MO, USA);
- Calcium chloride—$CaCl_2$ (Merck KGaA, Darmstadt, Germany);
- Calcium chloride hexahydrate—$CaCl_2(H_2O)_6$ (Sigma, St. Louis, MO, USA);
- Defibrinated sheep blood (Oxoid Limited, Basingstoke, UK);
- DMSO (Sigma-Aldrich, St. Louis, MO, USA);
- Dipotassium hydrogen phosphate—$K_2HPO_4$ (Honeywell Fluka, Seelze, Germany);
- DL-lactic acid (Sigma, St. Louis, MO, USA);
- FOS from chicory root (Megazyme, Bray, Ireland);
- Glycerol—analytic grade (Fisher Scientific, Loughborough, UK);
- GRS Universal Ladder (Grisp, Porto, Portugal);
- Microbial DNA-free water (Quiagen, Hilden, Germany);
- Hemin (Sigma, St. Louis, MO, USA);
- Hydrochloric acid—HCl (Honeywell Fluka, Seelze, Germany);
- L-cysteine HCl (Sigma-Aldrich, St. Louis, MO, USA);
- Magnesium sulfate—$MgSO_4$ (Honeywell Fluka, Seelze, Germany);
- Magnesium sulfate heptahydrate—$MgSO_4(H_2O)_7$ (Sigma, St. Louis, MO, USA);
- NZYSpeedy qPCR Green Master Mix ($2\times$) (Nzytech, Lisbon, Portugal);
- Pancreatin from the porcine pancreas—P7545 (Sigma, St. Louis, MO, USA);
- Pepsin from porcine gastric mucose powder—P7000 (Sigma, St. Louis, MO, USA);
- Peptone from animal tissue (Sigma, St. Louis, MO, USA);
- Phosphate buffered saline (Dulbecco A)—PBS (Oxoid Limited, Basingstoke, UK);

- Potassium dihydrogen phosphat—eKH$_2$PO$_4$ (Merck KGaA, Darmstadt, Germany);
- Propionic acid (Sigma, St. Louis, MO, USA);
- Resazurin sodium salt (Sigma, St. Louis, MO, USA);
- Sodium chloride—NaCl (Honeywell Fluka, Seelze, Germany);
- Sodium hydrogen carbonate—NaHCO$_3$ (Panreac, Barcelona, Spain);
- Sodium hydroxide—NaOH (LabChem, Zelienople, USA);
- Soja 115 INT2—complete feed for broiler chickens (Sorgal S.A., Aveiro, Portugal);
- Sulfuric acid—H$_2$SO$_4$ (Honeywell Fluka, Seelze, Germany);
- TAE Buffer 50× solution—RNAse free solution (Nzytech, Lisbon, Portugal);
- Tween 80 (Sigma, St. Louis, MO, USA);
- Vitamin K1 (Sigma, St. Louis, MO, USA);
- Yeast extract (Sigma, St. Louis, MO, USA).

*Appendix A.2. Culture Media Used in This Study*

- Columbia agar base—CBA (Liofilchem, Roseto degli Abruzzi, Italy);
- de Man, Rogosa, and Sharpe agar—MRSA (Biokar Diagnostics, Allonne, France);
- MacConkey agar—MCA (Biolife, Milan, Italy);
- Mueller–Hinton broth—MHB (Biokar Diagnostics, Allonne, France);
- Tryptic soy broth—TSB (Biokar Diagnostics, Allonne, France).

*Appendix A.3. Apparatus Used in This Study*

- 1 kDa molecular weight cut-off regenerated cellulose dialysis tubing Spectra/Por® 6 (Spectrum, NB, USA);
- Agilent 1260 II series HPLC (Agilent, Santa Clara, CA, USA);
- Alpha 2-4 LSC plus model (Martin Christ Gefriertrocknungsanlagen GmbH, Osterode am Harz, Germany)
- Anaerobic cabinet, Whitley A35 workstation (Don Whitley Scientific, Bingley, UK);
- Biometra Compact Multi-Wide (Analytik-Jena, Jena, Germany);
- Clifton NE4-22P digital water circulator (Nickel-Electro Ltd., Weston-super-Mare, UK);
- CryoCube® F570n freezer (Eppendorf, Hamburg, Germany);
- FerMac 260 pH controller (Electrolab Biotech Ltd., Gloucestershire, UK);
- Ion-exclusion Aminex HPX-87H column (Biorad, Hercules, CA, USA);
- Heraeus™ Megafuge™ 16R Centrifuge (Thermo Fischer Scientific, Waltham, MA, USA);
- Mixwel® laboratory blender (Alliance Bio Expertise Guipry, France);
- MR Hei-Tec magnetic stirrer (Heidolph Instruments GmbH & CO. KG, Schwabach, Germany);
- MST magnetic stirrer (Velp Scientifica, Usmate Velate, Italy);
- Oxoid™ AnaeroGen™ 2.5 L sachet (Thermo Fischer Scientific, Waltham, MA, USA);
- Oxoid™ AnaeroJar™ 2.5 L (Thermo Fischer Scientific, Waltham, MA, USA);
- pH sensor 405-DPAS-SC-K8S/200 (Mettler Toledo, Urdorf, Switzerland);
- PureLink™ Microbiome DNA Purification Kit (Thermo Fisher Scientific, Waltham, MA, USA);
- qTower$^3$ G (Analytik-Jena, Jena, Germany);
- Qubit 4 fluorometer (Thermo Fisher Scientific, Waltham, MA, USA);
- Qubit® dsDNA HS assay Kit (Thermo Fisher Scientific, Waltham, MA, USA);
- Reax top vortex (Heidolph Instruments GmbH & CO. KG, Schwabach, Germany);
- Refrigerator Beko RSNE445E33WN (Beko, Istanbul, Turkey);
- Sension+ 9663 ammonium ion selective electrode (ISE) (Hach, Loveland, CO, USA);
- SevenCompact pH meter (Mettler Toledo, Urdorf, Switzerland);
- SW22 shaking water bath (Julabo GmbH, Seelbach, Germany);
- Tamper-proof specimen 1-L containers (Sigma, St. Louis, MO, USA);
- UVP ChemStudio imagers (Analytik-Jena, Jena, Germany).

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
