# Peer review of "Development of a Chicken Gastrointestinal Tract (GIT) Simulation Model: Impact of Cecal Inoculum Storage Preservation Conditions"

_2673-8007, doi:10.3390/applmicrobiol3030066_

Round 1

Reviewer 1 Report

- The manuscript has merit and deals with a subject of high interest.

- English need to modify. And the sentences are too long, confusing, and complicated to read. It needs to be clearer and more specific.

- There are comments and inquiries that appear on the PDF file.

Reviewer 2 Report

This paper predicted the effects of dietary additives on chicken intestinal flora by building a chicken gastrointestinal tract (GIT) simulation model, the experimental design and methodology were feasible. However, the differences that existed between the in vitro simulation model and the experimental animals could not be assessed. Therefore, it would be more appropriate for the authors to discuss it in paper to show the significance of this research.

Abbreviations are required to be used in their full form when they first appear, but they are repeated in many places in the paper. Authors are requested to double-check and revise to the correct format.

Introduction

L59-60, Reference citations are needed here.

L63-64, Reference citations are needed here.

L77-78, Reference citations are needed here.

Result

L310-311, It might be inadequate for this conclusion, as the trend described by the authors cannot be clearly seen in the figure.

Disussion

The discussion section was informative but too long. It is suggested that the author could refine these arguments, which might be more logical.

L464-489, There is an abundance of descriptions of experimental results, but discussion of the role played by bacterial viability during cecal fermentation has been neglected.

Figure

Figure 1, The figure is not clear enough.

Figure 4-7, The x-axis and y-axis numbers can be enlarged or bolded. And the way of letters are labelled in Figures 5 and 6 is particularly confusing, it should be make adjustments. Perhaps the letters would be appropriate above the error line.

Table

Table 3, The format of the headings should be harmonised with the other tables.

Reviewer 3 Report

There are many repetitive sentences across the manuscript and needs to be largely reworked. 

Introduction

The main objective should be coupled with the hypothesis of the study and should be clearly stated in the introduction.

Material and Methods

The methods information should be adequately provided. Please modify the Ethical information, as the date of approval is missing (the project identification code, date of approval, and name of the ethics committee or institutional review board should be stated in the Materials and Methods section).  To facilitate the reader's comprehension of the experimental procedure, it is imperative to include a clear statement elucidating the experimental design at the outset of the Materials and Methods section. The layout of the methodology is mystifying. I would strongly suggest to the author to consider the following subheading for each Phase/experiment: Study design & The objective of the experiment.

The authors did not describe/justify selection criteria of broiler chickens obtained? As the gut microbiota can be dependent on the environment, type of diet given, housing conditions, etc.? Additionally, the variation between individual birds should be assessed, or at least taken into consideration. Also, the human handlers can affect the chicken microbiota. These effects must be excluded before considering any significant difference.

 Discussion/Conclusion

Certainly, discussing the limitations and constraints of the study and proposing future work are essential components that should be included in the discussion section of the study. These sections help provide a comprehensive understanding of the study's scope and potential areas for improvement. The statistical analysis is an important part of the work, it needs to be clearly summarized and the main results as key messages could be summarized in the conclusion for a better understanding.

Round 2

Reviewer 1 Report

The corrections are satisfactory for me and the paper can be published in present form. 

Reviewer 3 Report

The authors responded adequately to the raised points. The manuscript has been significantly improved and now permits publication in Applied Microbiology.